# Longitudinal high-throughput TCR repertoire profiling reveals the dynamics of T-cell memory formation after mild COVID-19 infection

Anastasia A Minervina[1], Ekaterina A Komech[1,2], Aleksei Titov[3], Meriem Bensouda Koraichi[4], Elisa Rosati[5], Ilgar Z Mamedov[1,2,6,7], Andre Franke[5], Grigory A Efimov[3], Dmitriy M Chudakov[1,2,6], Thierry Mora[4], Aleksandra M Walczak[4], Yuri B Lebedev[1,8], Mikhail V Pogorelyy[1,2]*

[1]Shemyakin-Ovchinnikov Institute of Bioorganic Chemistry, Moscow, Russian Federation; [2]Pirogov Russian National Research Medical University, Moscow, Russian Federation; [3]National Research Center for Hematology, Moscow, Russian Federation; [4]Laboratoire de physique de l'École Normale Supérieure, ENS, PSL, Sorbonne Universite, Universite de Paris, and CNRS, Paris, France; [5]Institute of Clinical Molecular Biology, Kiel University, Kiel, Germany; [6]Masaryk University, Central European Institute of Technology, Brno, Czech Republic; [7]V.I. Kulakov National Medical Research Center for Obstetrics, Gynecology and Perinatology, Moscow, Russian Federation; [8]Moscow State University, Moscow, Russian Federation

**Abstract** COVID-19 is a global pandemic caused by the SARS-CoV-2 coronavirus. T cells play a key role in the adaptive antiviral immune response by killing infected cells and facilitating the selection of virus-specific antibodies. However, neither the dynamics and cross-reactivity of the SARS-CoV-2-specific T-cell response nor the diversity of resulting immune memory is well understood. In this study, we use longitudinal high-throughput T-cell receptor (TCR) sequencing to track changes in the T-cell repertoire following two mild cases of COVID-19. In both donors, we identified CD4[+] and CD8[+] T-cell clones with transient clonal expansion after infection. We describe characteristic motifs in TCR sequences of COVID-19-reactive clones and show preferential occurrence of these motifs in publicly available large dataset of repertoires from COVID-19 patients. We show that in both donors, the majority of infection-reactive clonotypes acquire memory phenotypes. Certain T-cell clones were detected in the memory fraction at the pre-infection time point, suggesting participation of pre-existing cross-reactive memory T cells in the immune response to SARS-CoV-2.

*For correspondence:
m.pogorely@gmail.com

Competing interest: See
page 12

Reviewing editor: Sandeep
Krishna, National Centre for
Biological Sciences-Tata Institute
of Fundamental Research, India

## Introduction

COVID-19 is a global pandemic caused by the novel SARS-CoV-2 betacoronavirus (*Vabret et al., 2020*). T cells are crucial for clearing respiratory viral infections and providing long-term immune memory (*Schmidt and Varga, 2018*; *Swain et al., 2012*). Two major subsets of T cells participate in the immune response to viral infection in different ways: activated CD8[+] T cells directly kill infected cells, while subpopulations of CD4[+] T cells produce signalling molecules that regulate myeloid cell behaviour, drive, and support CD8 response and the formation of long-term CD8 memory, and participate in the selection and affinity maturation of antigen-specific B cells, ultimately leading to the

generation of neutralizing antibodies. In SARS-1 survivors, antigen-specific memory T cells were detected up to 11 years after the initial infection, when viral-specific antibodies were undetectable (*Ng et al., 2016*; *Oh et al., 2011*). The T-cell response was shown to be critical for protection in SARS-1-infected mice (*Zhao et al., 2010*). Patients with X-linked agammaglobulinemia, a genetic disorder associated with lack of B cells, have been reported to recover from symptomatic COVID-19 (*Quinti et al., 2020*; *Soresina et al., 2020*), suggesting that in some cases, T cells are sufficient for viral clearance. Thevarajan et al. showed that activated CD8$^+$HLA$^-$DR$^+$CD38$^+$ T cells in a mild case of COVID-19 significantly expand following symptom onset, reaching their peak frequency of 12% of CD8$^+$ T cells on day 9 after symptom onset, and contract thereafter (*Thevarajan et al., 2020*). Given the average time of 5 days from infection to the onset of symptoms (*Bi et al., 2020*), the dynamics and magnitude of T-cell response to SARS-CoV-2 is similar to that observed after immunization with live vaccines (*Miller et al., 2008*). SARS-CoV-2-specific T cells were detected in COVID-19 survivors by activation following stimulation with SARS-CoV-2 proteins (*Ni et al., 2020*) or by viral protein-derived peptide pools (*Weiskopf et al., 2020*; *Braun et al., 2020*; *Snyder et al., 2020*; *Le Bert et al., 2020*; *Meckiff et al., 2020*; *Bacher et al., 2020*; *Peng et al., 2020*). Some of the T cells activated by peptide stimulation were shown to have a memory phenotype (*Weiskopf et al., 2020*; *Le Bert et al., 2020*; *Mateus et al., 2020*), and some potentially cross-reactive CD4$^+$ T cells were found in healthy donors (*Braun et al., 2020*; *Grifoni et al., 2020*; *Sekine et al., 2020*; *Bacher et al., 2020*).

T cells recognise short pathogen-derived peptides presented on the cell surface of the major histocompatibility complex (MHC) using hypervariable T-cell receptors (TCR). TCR repertoire sequencing allows for the quantitative tracking of T-cell clones in time, as they go through the expansion and contraction phases of the response. It was previously shown that quantitative longitudinal TCR sequencing is able to identify antigen-specific expanding and contracting T cells in response to yellow fever vaccination with high sensitivity and specificity (*Minervina et al., 2020*; *Pogorelyy et al., 2018*; *DeWitt et al., 2015*). Not only clonal expansion but also significant contraction from the peak of the response are distinctive traits of T-cell clones specific to the virus (*Pogorelyy et al., 2018*).

In this study, we use longitudinal TCRalpha and TCRbeta repertoire sequencing to quantitatively track T-cell clones that significantly expand and contract after recovery from a mild COVID-19 infection, and determine their phenotype. We reveal the dynamics and the phenotype of the memory cells formed after infection, identify pre-existing T-cell memory clones participating in the response, and describe public TCR sequence motifs of SARS-CoV-2-reactive clones, suggesting a response to immunodominant epitopes.

## Results

### Longitudinal tracking of TCR repertoires of COVID-19 patients

In the middle of March (day 0), donor W female and donor M male (both healthy young adults) returned to their home country from the one of the centres of the COVID-19 outbreak in Europe at the time. Upon arrival, according to local regulations, they were put into strict self-quarantine for 14 days. On day 3 of self-isolation, both developed low-grade fever, fatigue, and myalgia, which lasted 4 days and was followed by a temporary loss of smell for donor M. On days 15, 30, 37, 45, and 85, we collected peripheral blood samples from both donors (*Figure 1a*). The presence of IgG and IgM SARS-CoV-2-specific antibodies in the plasma was measured at all time points using SARS-CoV-2 S-RBD domain-specific ELISA (*Figure 1—figure supplement 1*). From each blood sample, we isolated peripheral blood mononuclear cells (PBMCs, in two biological replicates), CD4$^+$, and CD8$^+$ T cells. Additionally, on days 30, 45, and 85, we isolated four T-cell memory subpopulations (*Figure 1—figure supplement 2a*): effector memory (EM: CCR7$^-$CD45RA$^-$), effector memory with CD45RA re-expression (EMRA: CCR7$^-$CD45RA$^+$), central memory (CM: CCR7$^+$CD45RA$^-$), and stem cell-like memory (SCM: CCR7$^+$CD45RA$^+$CD95$^+$). From all samples, we isolated RNA and performed TCRalpha and TCRbeta repertoire sequencing as previously described (*Pogorelyy et al., 2017*). For both donors, TCRalpha and TCRbeta repertoires were obtained for other projects 1 and 2 years prior to infection. Additionally, TCR repertoires of multiple samples for donor M – including sorted memory subpopulations – are available from a published longitudinal TCR sequencing study after yellow fever vaccination (donor M1 samples in *Minervina et al., 2020*).

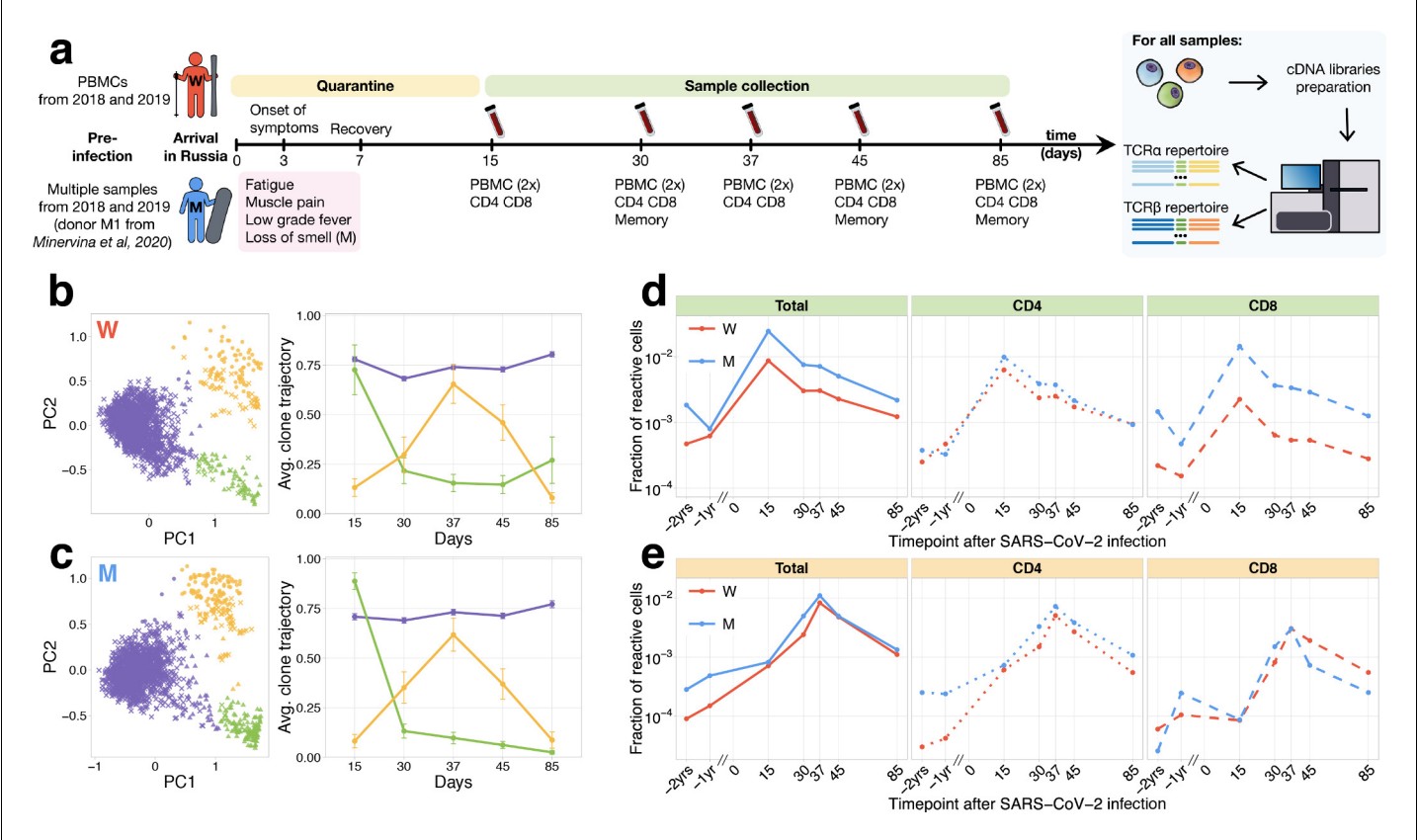

**Figure 1.** Longitudinal tracking of T-cell clones after mild COVID-19. (**a**) Study design. Peripheral blood of two donors was sampled longitudinally on days 15, 30, 37, 45, and 85 after arrival in Russia. At each time point, we evaluated SARS-CoV-2-specific antibodies using ELISA (***Figure 1—figure supplement 1***) and isolated PBMCs in two biological replicates. Additionally, CD4[+] and CD8[+] cells were isolated from a separate portion of blood, and EM, CM, EMRA, and SCM memory subpopulations were FACS sorted on days 30, 45, and 85. For each sample, we sequenced TCRalpha and TCRbeta repertoires. For both donors, pre-infection PBMC repertoires were sampled in 2018 and 2019 for other projects. (**b, c**) PCA of clonal temporal trajectories identifies three groups of clones with distinctive behaviours. Left: First two principal components of the 1000 most abundant TCRbeta clonotype frequencies normalized by maximum value for each clonotype in PBMC at post-infection time points. Colour indicates hierarchical clustering results of principal components; symbol indicates if clonotype was called as significantly contracted from day 15 to day 85 (triangles) or expanded from day 15 to day 37 (circles) by both edgeR and NoisET (***Figure 1—figure supplement 5*** shows overlap between clonal trajectory clusters and edgeR/NoisET hits). Right: Each curve shows the average ±2.96 SE of normalized clonal frequencies from each cluster. Contracting (**d**) and expanding (**e**) clones include both CD4[+] and CD8[+] T cells and are less abundant in pre-infection repertoires. T-cell clones significantly contracted from day 15 to day 85 (**d**) and significantly expanded from day 15 to day 37 (**e**) were identified in both donors. The fraction of contracting (**d**) and expanding (**e**) TCRbeta clonotypes in the total repertoire (calculated as the sum of frequencies of these clonotypes in the second PBMC replicate at a given time point and corresponding to the fraction of responding cells of all T cells) is plotted in log-scale for all reactive clones (left) and reactive clones with the CD4 (middle) and CD8 (right) phenotypes. Similar dynamics were observed in TCRalpha repertoires (***Figure 1—figure supplement 3***) and for significantly expanded/contracted clones identified with the NoisET Bayesian differential expansion statistical model alone (***Figure 1—figure supplement 4***).

The online version of this article includes the following figure supplement(s) for figure 1:

**Figure supplement 1.** Both donors developed anti-SARS-CoV-2 IgG and IgM responses by day 15 post-infection.

**Figure supplement 2.** Gating strategy for T cell subsets.

**Figure supplement 3.** Longitudinal tracking of T-cell clones after mild COVID-19 with TCRalpha repertoire sequencing.

**Figure supplement 4.** Comparison of edgeR and NoisET clonal expansion detection procedures.

**Figure supplement 5.** The overlap between clusters of clonal trajectories identified by PCA and groups of expanding/contracting clones identified with edgeR/NoisET.

## Two waves of T-cell clone response

From previously described activated T-cell dynamics for SARS-CoV-2 (***Thevarajan et al., 2020***), and immunization with live vaccines (***Miller et al., 2008***), the peak of the T-cell expansion is expected around day 15 post-infection, and responding T cells significantly contract afterwards. However,

*Weiskopf et al., 2020* reports an increase of SARS-CoV-2-reactive T cells at later time points, peaking in some donors after 30 days following symptom onset. To identify groups of T-cell clones with similar dynamics in an unbiased way, we used principal component analysis (PCA) in the space of T-cell clonal trajectories (*Figure 1b,c*). This exploratory data analysis method allows us to visualize major trends in the dynamics of abundant TCR clonotypes (occurring within top 1000 on any post-infection time points) between multiple time points.

In both donors, and in both TCRalpha and TCRbeta repertoires, we identified three clusters of clones with distinct dynamics. The first cluster (*Figure 1b,c*, purple) corresponded to abundant TCR clonotypes that had constant concentrations across time points, the second cluster (*Figure 1b,c*, green) showed contraction dynamics from day 15 to day 85, and the third cluster (*Figure 1b,c*, yellow) showed an unexpected clonal expansion from day 15 with a peak on day 37 followed by contraction. The clustering and dynamics are similar in both donors and are reproduced in TCRbeta (*Figure 1b,c*) and TCRalpha (*Figure 1—figure supplement 3a,b*) repertoires. We next used edgeR, a software for differential gene expression analysis (*Robinson et al., 2010*) and NoisET, a Bayesian differential expansion model (*Puelma Touzel et al., 2020*), to specifically detect changes in clonotype concentration between pairs of time points in a statistically reliable way and without limiting the analysis to the most abundant clonotypes. Both NoisET and edgeR use biological replicate samples collected at each time point to train a noise model for sequence counts. Results for the two models were similar (*Figure 1—figure supplement 4*), and we conservatively defined as expanded or contracted the clonotypes that were called by both models simultaneously. We identified 291 TCRalpha and 295 TCRbeta clonotypes in donor W and 607 TCRalpha and 616 TCRbeta in donor M significantly contracted from day 15 to day 85 (largely overlapping with cluster 2 of clonal trajectories, *Figure 1—figure supplement 5*). One hundred and seventy six TCRalpha and 278 TCRbeta for donor W and 293 TCRalpha and 427 TCRbeta clonotypes for donor M were significantly expanded from days 15 to 37 (corresponding to cluster 3 of clonal trajectories).

Note that, to identify putatively SARS-CoV-2-reactive clones, we only used post-infection time points, so that our analysis can be reproduced in other patients and studies where pre-infection time points are unavailable. However, tracking the identified responding clones back to pre-infection time points reveals strong clonal expansions from pre- to post-infection (*Figure 1d,e*, *Figure 1—figure supplement 3c,d*). For brevity, we further refer to clonotypes significantly contracted from day 15 to 85 as *contracting* clones and clonotypes significantly expanding from day 15 to 37 as *expanding* clones. Contracting clones corresponded to 2.5% and 0.9% of T cells on day 15 post-infection, expanding clones reached 1.1% and 0.8% on day 37 for donors M and W, respectively (*Figure 1d,e*, left). This magnitude of the T-cell response is of the same order of magnitude as previously observed after live yellow fever vaccine immunization of donor M (6.7% T cells on day 15 post-vaccination). For each contracting and expanding clone, we determined their CD4/CD8 phenotype using separately sequenced repertoires of CD4$^+$ and CD8$^+$ subpopulations (see Materials and methods). Both CD4$^+$ and CD8$^+$ subsets participated actively in the response (*Figure 1d,e*). Interestingly, clonotypes expanding after day 15 were significantly biased towards the CD4$^+$ phenotype, while contracting clones had balanced CD4/CD8 phenotype fractions in both donors (Fisher's exact test, $p<0.01$ for both donors). A recent study by *Parrot et al., 2020* reported activation of mucosal-associated invariant T (MAIT) cells in COVID-19 patients. MAIT cells are innate-like T cells recognising bacterial metabolites with TCRs featuring an invariant TCRalpha chain. We used a set of 121 unique MAIT TCRalpha amino acid sequences from *Loh et al., 2020* to check whether MAIT cells contributed significantly to contracting or expanding clonotype groups identified in our study. We found that a small number of low-frequency contracting (one clonotype for each donor) and expanding (two clonotypes for each donor) TCRalpha clonotypes have MAIT invariant alpha chains, and the contribution of these clonotypes to the response was relatively small (<1% of responding T cells are MAIT).

## Memory formation and pre-existing memory

On days 30, 45, and 85, we identified both contracting (*Figure 2a–c*) and expanding (*Figure 2—figure supplement 1a–c*) T-cell clones in the memory subpopulations of peripheral blood. Both CD4$^+$ and CD8$^+$ responding clones were found in the CM and EM subsets; however, CD4$^+$ were more biased towards CM (with exception of donor W day 30 time point, where a considerable fraction of CD8$^+$ clones were found in CM), and CD8$^+$ clones more represented in the EMRA subset. A small number of both CD4$^+$ and CD8$^+$ responding clonotypes were also identified in the SCM

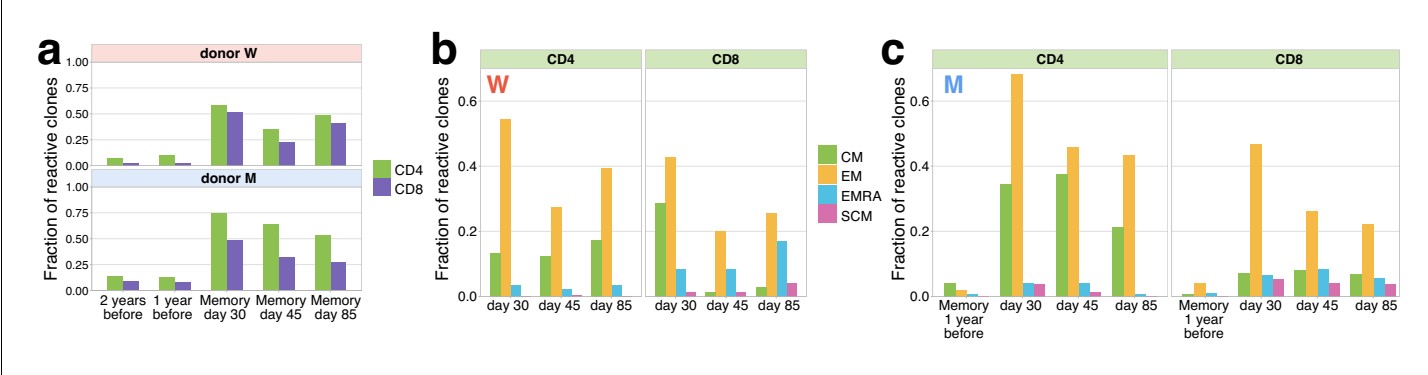

**Figure 2.** Memory phenotypes of responding clonotypes contracting after day 15. (a) A large fraction of contracting clonotypes is identified in T-cell memory subsets after infection. Bars show the fraction of contracting CD4+ and CD8+ TCRbeta clonotypes present in 2 year; 1 year pre-infection PBMC; in at least one of memory subpopulation sampled on day 30, day 37, and day 85 post-infection. (b, c) Responding clones are found in different memory subsets. Fraction of CD4+ (left panels) and CD8+ (right panels) contracting clones of donor W (b) and M (c) that were identified in each memory subpopulation repertoire at each time point. For both donors, CD4+ clonotypes were found predominantly in central memory (CM) and effector memory (EM), while CD8+ T cells were enriched in EMRA compartment. (c) For donor M, CD4+ contracting clonotypes are also identified in memory subsets 1 year before the infection, with a bias towards the CM subpopulation and a group of CD8+ clones is found in the pre-infection EM subpopulation.

The online version of this article includes the following figure supplement(s) for figure 2:

**Figure supplement 1.** Memory phenotypes of responding clonotypes expanding from day 15 to day 37.

**Figure supplement 2.** Dynamics of pre-existing SARS-CoV-2 responding clones.

subpopulation, which was previously shown to be a long-lived T-cell memory subset (*Fuertes Marraco et al., 2015*). Note that we sequenced more cells from PBMC than from the memory subpopulations (*Supplementary file 1*), so that some low-abundant responding T-cell clones are not sampled in the memory subpopulations. Intriguingly, a number of responding CD4+ clones, and fewer CD8+ clones, were also represented in the repertoires of both donors 1 and 2 years before the infection. The majority of the pre-existing clones had low concentrations at pre-infection and day 85 post-infection time points, and high concentrations on day 15, suggesting that they expanded after the infection and later contracted for both donors (*Figure 2—figure supplement 2*). For donor M, for whom we had previously sequenced memory subpopulations before the infection (*Minervina et al., 2020*), we were able to identify pre-existing SARS-CoV-2-reactive CD4+ clones in the CM subpopulation 1 year before the infection and a group of CD8+ clones in the pre-infection EM subpopulation. Interestingly, on day 30 after infection, the majority of pre-infection CM clones were detected in the EM subpopulation, suggesting recent T-cell activation and a switch of the phenotype from memory to effector. These clones might represent memory T cells cross-reactive for other infections, for example other human coronaviruses.

A search for TCRbeta amino acid sequences of responding clones in VDJdb (*Bagaev et al., 2020*) — a database of TCRs with known specificities — resulted in essentially no overlap with TCRs not specific for SARS-CoV-2 epitopes: only two clonotypes matched. One match corresponded to the cytomegalovirus epitope presented by the HLA-A*03 MHC allele, which is absent in both donors (*Supplementary file 2*), and a second match was for Influenza A virus epitope presented by HLA-A*02 allele. The absence of matches suggests that contracting and expanding clones are unlikely to be specific for immunodominant epitopes of common pathogens covered in VDJdb. We next asked if we could map specificites of our responding clones to SARS-CoV-2 epitopes.

## Validation by MHC tetramer-staining assay

On day 25, post-infection donor M participated in a study by *Shomuradova et al., 2020* (as donor p1434), where his CD8+ T cells were stained with a HLA-A*02:01-YLQPRTFLL MHC-I tetramer (*Figure 1—figure supplement 2b*). TCRalpha and TCRbeta of FACS-sorted tetramer-positive cells were sequenced and deposited to VDJdb (see *Shomuradova et al., 2020* for the experimental details). We matched these tetramer-specific TCR sequences to our longitudinal dataset (*Figure 3a* for

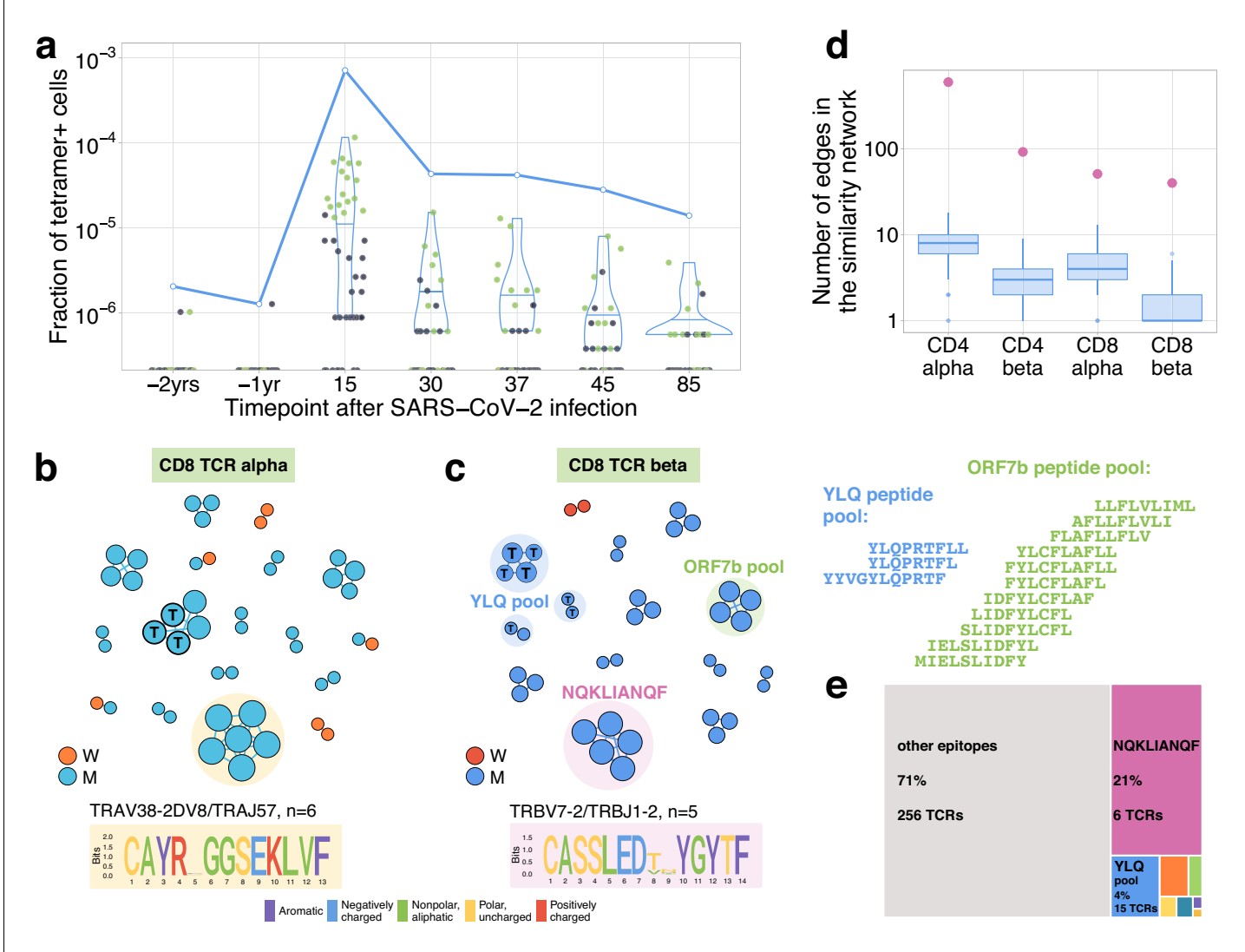

**Figure 3.** Distinctive TCR sequence motifs and epitope specificity of contracting CD8[+] T cell clones. (**a**) SARS-CoV-2-specific TCRs are independently identified by clonal contraction. On day 25, donor M participated in a study by *Shomuradova et al., 2020*, where TCR sequences responding to the HLA-A*02:01-YLQPRTFLL-tetramer T cells were determined. Here we matched the resulting tetramer-specific TCRbeta clonotypes to the longitudinal repertoire dataset obtained in the current study. Each dot corresponds to the frequency of HLA-A*02:01-YLQPRTFLL-tetramer-specific TCRbeta clonotype in bulk repertoire of donor M (donor p1434 from *Shomuradova et al., 2020*) at each time point. Green dots correspond to clonotypes independently identified as contracting in our longitudinal dataset. Blue line shows the cumulative frequency of tetramer-specific TCRbeta clonotypes. (**b, c**) Analysis of TCR amino acid sequences of contracting CD8[+] clones reveals distinctive motifs. For each set of CD8alpha and CD8beta contracted clonotypes, we constructed a separate similarity network. Each vertex in the similarity network corresponds to a contracting clonotype. An edge indicates two or less amino acid mismatches in the CDR3 region and identical V and J segments. Networks are plotted separately for CD8alpha (**b**) and CD8beta (**c**) contracting clonotypes. Clonotypes without neighbours are not shown. Sequence logos corresponding to the largest clusters are shown under the corresponding network plots. 'T' on vertices indicate TCRbeta clonotypes confirmed by HLA-A*02:01-YLQPRTFLL tetramer staining. Shaded coloured circles (**c**) indicate clonotypes with large number of matches to CD8[+] TCRs recognising SARS-CoV-2 peptides pools from *Snyder et al., 2020* (MIRA peptide dataset). Lists of peptides in YLQ and ORF7b peptide pools are shown on the right. (**d**) Sequence convergence among contracting clonotypes. The number of edges in each group is shown by pink dots and is compared to the distribution of that number in 1000 random samples of the same size from the relevant repertoires at day 15 (blue boxplots). (**e**) Fraction of TCRbeta clonotypes with matches in the MIRA dataset (coloured rectangles) out of all responding CD8[+] TCRbeta clonotypes in donor M on day 15.

The online version of this article includes the following figure supplement(s) for figure 3:

**Figure supplement 1.** HLA-A*02:01-YLQPRTFLL-specific TCRs are independently identified by clonal contraction.

**Figure supplement 2.** ALICE algorithm output for TCRalpha PBMC repertoire of donor M on day 15.

TCRbeta and *Figure 3—figure supplement 1* for TCRalpha). We found that their frequencies were very low on pre-infection time points and monotonically decreased from their peak on day 15 ($7.1 \times 10^{-4}$ fraction of bulk TCRbeta repertoire) to day 85 ($1.3 \times 10^{-5}$ fraction), in close analogy to our contracting clone set. Among the tetramer-positive clones that were abundant on day 15 (with bulk frequency $>10^{-5}$), 17 of 18 TCRbetas and 12 of 15 TCRalphas were independently identified as contracting by our method (as expected for TCRs specifically binding MHC-I multimer, all matching clonotypes in our datasets had a CD8$^+$ phenotype). No tetramer-positive clonotypes matched with expanding clonotypes peaking on day 37.

## TCR sequence motifs of responding clones

It was previously shown that TCRs recognising the same antigens frequently have highly similar TCR sequences (*Dash et al., 2017*; *Glanville et al., 2017*). To identify motifs in TCR amino acid sequences, we plotted similarity networks for significantly contracted (*Figures 3b,c* and *4a,b*) and expanded (*Figure 4—figure supplement 1b–e*) clonotypes. The number of edges in all similarity networks except CD8$^+$ expanding clones was significantly larger than would expected by randomly

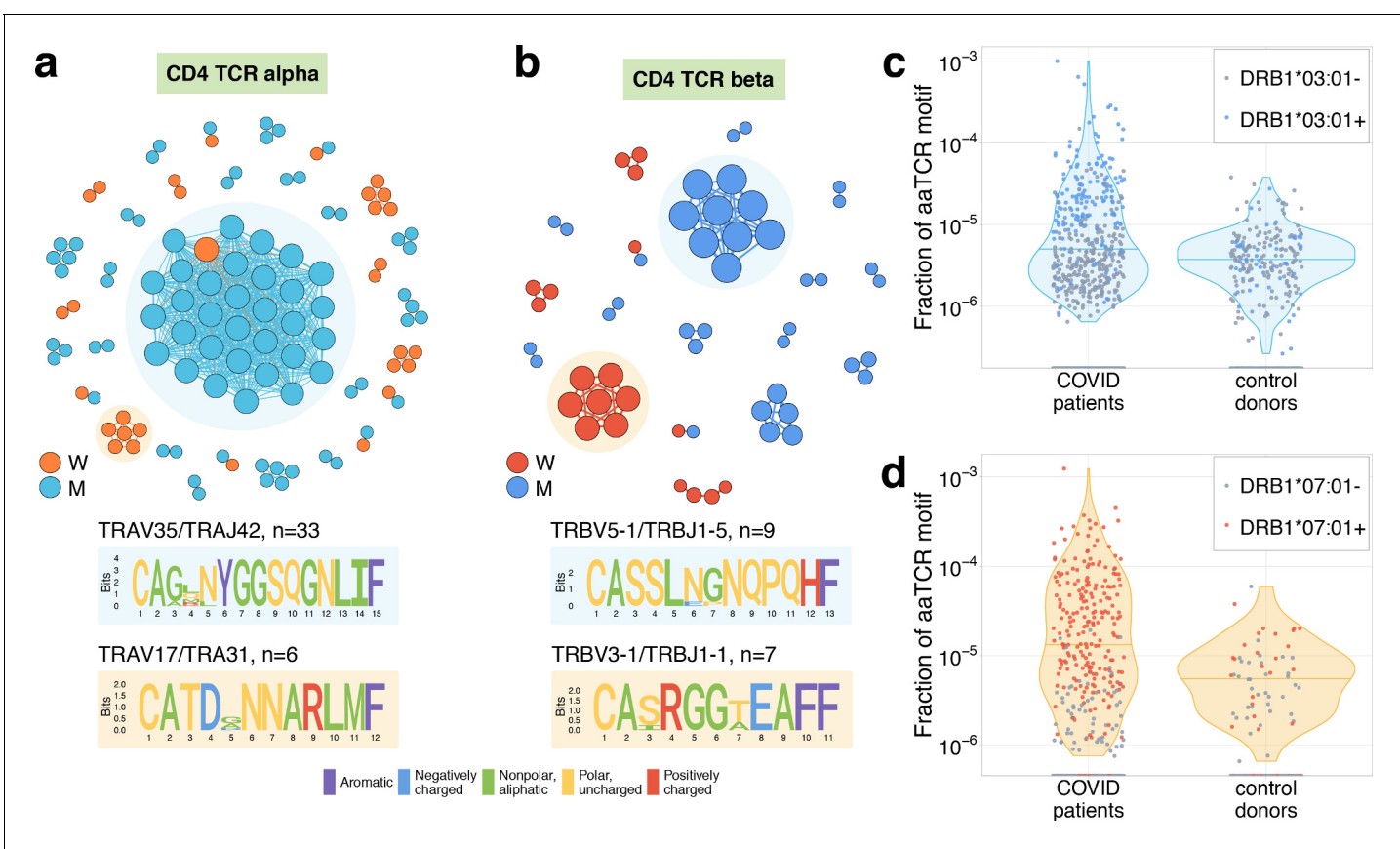

**Figure 4.** Analysis of TCR amino acid sequences of CD4$^+$ contracting clones reveals distinctive motifs. (a) Each vertex in the similarity network corresponds to a contracting clonotype. An edge indicates two or less amino acid mismatches in the CDR3 region (and identical V and J segments). Networks are plotted separately for CD4alpha (a) and CD4beta (b) contracting clonotypes. Clonotypes without neighbours are not shown. Sequence logos corresponding to the largest clusters are shown under the corresponding network plots. (c, d) Clonotypes forming the two largest motifs are significantly more clonally expanded (p<0.001, one-sided t-test) in a cohort of COVID-19 patients (*Snyder et al., 2020*) than in a cohort of control donors (*Emerson et al., 2017*). Each dot corresponds to the total frequency of clonotypes from motifs shaded on (b) in the TCRbeta repertoire of a given donor. Coloured dots show donors predicted to share HLA-DRB1*07:01 allele with donor W (c) or HLA-DRB1*03:01-DQB1*02:01 haplotype with donor M (d).

The online version of this article includes the following figure supplement(s) for figure 4:

**Figure supplement 1.** Expanding CD4$^+$ (but not CD8$^+$) clonotypes show unexpected TCRalpha and TCRbeta sequence convergence.

**Figure supplement 2.** Identification of COVID-19 patients by frequency of TCR motifs from contracting CD4$^+$ clones from donors M (a) and W (b).

sampling the same number of clonotypes from the corresponding repertoire (*Figure 3d* and *Figure 4—figure supplement 1a*). In both donors, we found clusters of highly similar clones in both CD4$^+$ and CD8$^+$ subsets for expanding and contracting clonotypes. Clusters were largely donor specific, as expected, since our donors have dissimilar HLA alleles (*Supplementary file 2*) and thus each is likely to present a non-overlapping set of T-cell antigens. The largest cluster, described by the motif TRAV35-CAGXNYGGSQGNLIF-TRAJ42, was identified in donor M's CD4$^+$-contracting alpha chains. Clones from this cluster constituted 16.3% of all of donor M's CD4$^+$-responding cells on day 15, suggesting a response to an immunodominant CD4$^+$ epitope in the SARS-CoV-2 proteome. The high similarity of the TCR sequences of responding clones in this cluster allowed us to independently identify motifs from donor M's CD4 alpha contracting clones using the ALICE algorithm (*Pogorelyy et al., 2019*; *Figure 3—figure supplement 2*). While the time-dependent methods (*Figure 1*) identify abundant clones, the ALICE approach is complementary to both edgeR and NoisET as it identifies clusters of T cells with similar sequences independent of their individual abundances.

## Mapping TCR motifs to SARS-CoV-2 epitopes

In CD8$^+$ T cells, three clusters of highly similar TCRbeta clonotypes in donor M and one cluster of TCRalpha clonotypes correspond to YLQPRTFLL-tetramer-specific TCR sequences described above. To map additional specificities for CD8$^+$ TCRbetas, we used a large set of SARS-CoV-2-peptide-specific TCRbeta sequences from *Snyder et al., 2020* obtained using Multiplex Identification of Antigen-specific T-cell Receptors Assay (MIRA) with combinatorial peptide pools (*Klinger et al., 2015*). For each responding CD8$^+$ TCRbeta, we searched for the identical or highly similar (same VJ combination, up to one mismatch in CDR3aa) TCRbeta sequences specific for given SARS-CoV-2 peptides. A TCRbeta sequence from our set was considered mapped to a given peptide if it had at least two highly similar TCRbeta sequences specific for this peptide in the MIRA experiment. This procedure yielded unambiguous matches for 32 CD8$^+$ TCRbetas – just one clonotype was paired to two peptide pools (*Supplementary file 3*). The vast majority of matches to MIRA corresponded to groups of contracting clones. As expected, we found that all clusters corresponding to HLA-A*02:01-YLQPRTFLL MHC-I tetramer-specific TCRs were matched to the peptide pool YLQPRTFL, YLQPRTFLL,YVVGYLQPRTF in the MIRA dataset. Another large group of matches corresponded to the HLA-B*15:01-restricted (*Stamatakis et al., 2020*) NQKLIANQF epitope. Interestingly, clonotypes corresponging to this cluster together made up 21% of the CD8$^+$ immune response on day 15, suggesting immunodominance of this epitope. Two TCRbeta clonotypes mapped to this epitope were identified in Effector Memory subset 1 year before the infection, suggesting potential cross-reactive response. We speculate that this response might be initially triggered by NQKLIANAF, a homologous HLA-B*15:01 epitope from HKU1 or OC43, common human betacoronaviruses. Importantly, no matches for the NQKLIANQF epitope were observed among contracting or expanding clonotypes of HLA-B*15-negative donor W (*Supplementary file 2*). To predict potential pairings between TCRalpha and TCRbeta motifs, we used a method of alpha/beta clonal trajectory matching described in *Minervina et al., 2020* (see Materials and methods for details). We found consistent pairing between one of the motifs in TCRalpha to the largest motif in TCRbeta T cells, which is associated with HLA-B*15:01-NQKLIANQF.

## Validation of CD4$^+$ COVID-19 HLA-restricted specificity by cohort association analysis

At the time of writing, no data on TCR sequences specific to MHC-II class epitopes exist to map specificities of CD4$^+$ T cells in a similar way as we did with MIRA-specific TCRs. However, a recently published database of 1414 bulk TCRbeta repertoires from COVID-19 patients allowed us to confirm the SARS-CoV-2 specificity of contracting clones indirectly. Public TCRbeta sequences that can recognise SARS-CoV-2 epitopes are expected to be clonally expanded and thus sampled more frequently in the repertoires of COVID-19 patients than in control donors. In *Figure 4c,d*, we show that the total frequency of TCRbeta sequences forming the largest cluster in donor M (*Figure 4c*) and donor W (*Figure 4d*) is significantly larger in the COVID-19 cohort than in the healthy donor cohort from *Emerson et al., 2017*, suggesting antigen-dependent clonal expansion. We hypothesized that the difference between control and COVID-19 donors in motif abundance should be even larger if we

restrict the analysis to donors sharing the HLA allele that presents the epitope. Unfortunately, HLA-typing information is not yet available for the COVID-19 cohort. However, using sets of HLA-associated TCRbeta sequences from *DeWitt et al., 2018*, we could build a simple classifier to predict the HLA alleles of donors from both the control and COVID-19 cohorts exploiting the presence of TCRbeta sequences associated with certain HLA alleles (see Materials and methods for details). We found that the CD4[+] TCRbeta motif from donor W occurs preferentially in donors predicted to have DRB1*07:01 allele, while the motif from donor M appears to be associated with HLA-DRB1*03:01-DQB1*02:01 haplotype. The frequency of sequences corresponding to these motifs can then be used to identify SARS-CoV-2 infected donors with matching HLA alleles (*Figure 4—figure supplement 2*).

## Discussion

Using longitudinal repertoire sequencing, we identified a group of CD4[+] and CD8[+] T cell clones that contract after recovery from a SARS-CoV-2 infection. Our response timelines agree with T-cell dynamics reported by *Thevarajan et al., 2020* for mild COVID-19, as well as with dynamics of T-cell response to live vaccines (*Miller et al., 2008*). We further mapped the specificities of contracting CD8[+] T cells using sequences of SARS-CoV-2-specific T cells identified with tetramer staining in the same donor, as well as the large set of SARS-CoV-2 peptide stimulated TCRbeta sequences from *Snyder et al., 2020*. For large CD4[+] TCRbeta motifs, we show strong association with COVID-19 by analysing the occurrence patterns and frequencies of these sequences in a large cohort of COVID-19 patients. Surprisingly, in both donors, we also identified a group of predominantly CD4[+] clonotypes that expanded from day 15 to day 37 after the infection. One possible explanation for this second wave of expansion is the priming of CD4[+] T cells by antigen-specific B cells, but there might be other mechanisms such as the migration of SARS-CoV-2-specific T cells from lymphoid organs or bystander activation of non-SARS-CoV-2 specific T cells. It is also possible that later expanding T cells are triggered by another infection, simultaneously and asymptomatically occurring in both donors around day 30. In contrast to the first wave of response identified by contracting clones, for now we do not have confirmation that this second wave of expansion corresponds to SARS-CoV-2-specific T cells. The accumulation of TCR sequences for CD4[+] SARS-CoV-2 epitope-specific T cells may further address this question. We showed that a large fraction of putatively SARS-CoV-2-reactive T cell clones are later found in memory subpopulations and remain there at least 3 months after infection. Importantly, some of responding clones are found in long-lived stem cell-like memory (SCM) subset, as also reported for SARS-CoV-2 convalescent patients in *Sekine et al., 2020*. A subset of CD4[+] clones were identified in pre-infection central memory subsets, and a subset of CD8[+] T cells were found in effector memory. Among these are CD8[+] clones recognising NQKLIANQF, an immunodominant HLA-B*15:01 restricted SARS-CoV-2 epitope, for which homologous epitope differing by 1 aa mismatch exists in common human betacoronaviruses. The presence of SARS-CoV-2 cross-reactive CD4[+] T cells in healthy individuals was recently demonstrated (*Braun et al., 2020*; *Grifoni et al., 2020*; *Le Bert et al., 2020*; *Meckiff et al., 2020*; *Bacher et al., 2020*; *Peng et al., 2020*). Our data further suggests that cross-reactive CD4[+] and CD8[+] T cells can participate in the response in vivo. It is interesting to ask whether the presence of cross-reactive T cells before infection is linked to the mildness of the disease (with predicted HLA-B*15:01 cross-reactive epitope described above as a good starting point). Larger studies with cohorts of severe and mild cases with pre-infection time points are needed to address this question.

## Materials and methods

### Key resources table

| Reagent type (species) or resource | Designation | Source or reference | Identifiers | Additional information |
|---|---|---|---|---|
| Antibody | Anti-CD3-FITC (mouse monoclonal) | eBioscience | CAT# 11-0038-42 | FACS (5 µl per test) |
| Antibody | Anti-CD45RA-eFluor450 (mouse monoclonal) | eBioscience | CAT# 48-0458-42 | FACS (5 µl per test) |
| Antibody | Anti-CCR7-APC (rat monoclonal) | eBioscience | CAT# 17-1979-42 | FACS (5 µl per test) |

*Continued on next page*

*Continued*

| Reagent type (species) or resource | Designation | Source or reference | Identifiers | Additional information |
|---|---|---|---|---|
| Antibody | Anti-CD95-PE (mouse monoclonal) | eBioscience | CAT# 12-0959-42 | FACS (5 µl per test) |
| Commercial assay or kit | Dynabeads CD4 Positive Isolation Kit | Invitrogen | CAT# 11331D | |
| Commercial assay or kit | Dynabeads CD8 Positive Isolation Kit | Invitrogen | CAT# 11333D | |

## Donors and blood samples

Peripheral blood samples from two young healthy adult volunteers, donor W (female) and donor M (male), were collected with written informed consent in a certified diagnostics laboratory. Both donors gave written informed consent to participate in the study under the declaration of Helsinki. HLA alleles of both donors (*Supplementary file 2*) were determined by an in-house cDNA high-throughput sequencing method.

## SARS-CoV-2 S-RBD domain-specific ELISA

An ELISA assay kit developed by the National Research Centre for Hematology was used for detection of anti-S-RBD IgG according to the manufacturer's protocol. The relative IgG level (OD/CO) was calculated by dividing the OD (optical density) values by the mean OD value of the cut-off positive control serum supplied with the Kit (CO). OD values of d37, d45, and d85 samples for donor M exceeded the limit of linearity for the Kit. In order to properly compare the relative IgG levels between d30, d37, d45, and d85, these samples were diluted 1:400 instead of 1:100; the ratios d37:d30 and d45:d30 and d85:d30 were calculated and used to calculate the relative IgG level of d37, d45, and d85 by multiplying d30 OD/CO value by the corresponding ratio. Relative anti-S-RBD IgM level was calculated using the same protocol with anti-human IgM-HRP-conjugated secondary antibody. Since the control cut-off serum for IgM was not available from the Kit, in *Figure 1—figure supplement 1b*, we show OD values for nine biobanked pre-pandemic serum samples from healthy donors.

## Isolation of PBMCs and T-cell subpopulations

PBMCs were isolated with the Ficoll-Paque density gradient centrifugation protocol. CD4$^+$ and CD8$^+$ T cells were isolated from PBMCs with Dynabeads CD4$^+$- and CD8$^+$-positive selection kits (Invitrogen), respectively. For isolation of EM, EMRA, CM, and SCM memory subpopulations, we stained PBMCs with the following antibody mix: anti-CD3-FITC (UCHT1, eBioscience), anti-CD45RA-eFluor450 (HI100, eBioscience), anti-CCR7-APC (3D12, eBioscience), and anti-CD95-PE (DX2, eBioscience). Cell sorting was performed on FACS Aria III, and all four isolated subpopulations were lysed with Trizol reagent immediately after sorting.

## TCR library preparation and sequencing

TCRalpha and TCRbeta cDNA libraries preparation was performed as previously described in *Pogorelyy et al., 2017*. RNA was isolated from each sample using Trizol reagent according to the manufacturer's instructions. A universal primer binding site, sample barcode, and unique molecular identifier (UMI) sequences were introduced using the 5′RACE technology with TCRalpha and TCRbeta constant segment-specific primers for cDNA synthesis. cDNA libraries were amplified in two PCR steps, with the introduction of the second sample barcode and Illumina TruSeq adapter sequences at the second PCR step. Libraries were sequenced using the Illumina NovaSeq platform (2 × 150 bp read length).

## TCR repertoire data analysis

### Raw data preprocessing

Raw sequencing data was demultiplexed, and UMI-guided consensuses were built using migec v.1.2.7 (*Shugay et al., 2014*). Resulting UMI consensuses were aligned to V and J genomic templates of the TRA and TRB locus and assembled into clonotypes with mixcr v.2.1.11 (*Bolotin et al., 2015*). See *Supplementary file 1* for the number of cells, UMIs, and unique clonotypes for each sample.

## Identification of clonotypes with active dynamics

Principal component analysis (PCA) of clonal trajectories was performed as described before *Minervina et al., 2020*. First, we selected clones that were present among the top 1000 abundant in any of post-infection PBMC repertoires, including biological replicates, i.e. considered clone abundant if it was found within top 1000 most abundant clonotypes in at least one of the replicate samples at one time point. Next, for each such abundant clone, we calculated its frequency at each post-infection time point and divided this frequency by the maximum frequency of this clone for normalization. Then we performed PCA on the resulting normalized clonal trajectory matrix and identified three clusters of trajectories with hierarchical clustering with average linkage, using Euclidean distances between trajectories. We identify statistically significant contractions and expansions with edgeR as previously described (*Pogorelyy et al., 2018*), using false discovery rate-adjusted $p<0.01$ and $\log_2$ fold-change threshold of 1. NoisET implements the Bayesian detection method described in *Puelma Touzel et al., 2020*. Briefly, a two-step noise model accounting for cell sampling and expression noise is inferred from replicates, and a second model of expansion is learned from the two time points to be compared. The procedure outputs the posterior probability of expansion or contraction and the median estimated $\log_2$ fold change, whose thresholds are set to 0.05 and 1, respectively.

## Mapping of COVID-19-associated TCRs to the MIRA database

TCRbeta sequences from T cells specific for SARS-CoV-2 peptide pools MIRA (ImmuneCODE release 2) were downloaded from https://clients.adaptivebiotech.com/pub/covid-2020. V and J genomic templates were aligned to TCR nucleotide sequences from the MIRA database using mixcr 2.1.11. We consider a TCRbeta from MIRA matched to a TCRbeta from our data, if it had the same V and J and at most one mismatch in CDR3 amino acid sequence. We consider a TCRbeta sequence mapped to an epitope if it has at least two identical or highly similar (same V, J and up to one mismatch in CDR3 amino acid sequence) TCRbeta clonotypes reactive for this epitope in the MIRA database.

## Computational alpha/beta pairing by clonal trajectories

Computational alpha/beta pairing was performed as described in *Minervina et al., 2020*. For each TCRbeta, we determine the TCRalpha with the closest clonal trajectory (*Supplementary files 3* and *5*). We observe no stringent pairings between TCRbeta and TCRbeta motifs with exception of two contracting CD8 TCRbeta clusters: TRBV7-2/TRBJ1-2 NQKLIANQF-associated clones from donor M paired to TRAV21/TRAJ40 alphas from the same cluster (CASSLEDTNYGYTF-CAVHSSGTYKYIF and CASSLEDTIYGYTF-CAALTSGTYKYIF), and TRBV7-9/TRBJ2-3 beta cluster paired to largest alpha cluster (CASSPTGRGRTDTQYF-CAYRSGGSEKLVF and CASSPTGRGGTDTQYF-CAYRRPGGEKLTF).

## Computational prediction of HLA types

To predict HLA types from TCR repertoires of COVID-19 cohort, we used sets of HLA-associated TCR sequences from *DeWitt et al., 2018*. We use TCRbeta repertoires of 666 donors from cohort from *Emerson et al., 2017*, for which HLA-typing information is available in *DeWitt et al., 2018* as a training set to fit logistic regression model, where the presence or absense of given HLA allele is an outcome, and the number of allele-associated sequences in repertoire, as well as the total number of unique sequences in the repertoire, are the predictors. A separate logistic regression model was fitted for each set of HLA-associated sequences from *DeWitt et al., 2018* and then used to predict the probability $p$ that a donor from the COVID-19 cohort has this allele. Donors with $p<0.2$ were considered negative for a given allele.

## Data availability

Raw sequencing data are deposited to the Short Read Archive (SRA) accession: PRJNA633317. Processed TCRalpha and TCRbeta repertoire datasets, resulting repertoires of SARS-CoV-2-reactive clones, and raw data preprocessing instructions can be accessed from: https://github.com/pogorely/Minervina_COVID; copy archived at https://archive.softwareheritage.org/swh:1:rev:d5b3953d3739f21a1ccc23114bfffa002e11951e/.

## Acknowledgements

The study was supported by RSF 20-15-00351. ER and AF are supported by the DFG Excellence Cluster Precision Medicine in Chronic Inflammation (Exc2167) and the DFG grant n. 4096610003, MBK, TM, and AMW are supported by European Research Council COG 724208. IZM is supported by RFBR 19-54-12011 and 18-29-09132. DMC is supported by the grant from Ministry of Science and Higher Education of the Russian Federation 075-15-2019-1789.

## Additional information

### Competing interests

Aleksandra M Walczak: Senior editor, *eLife*. The other authors declare that no competing interests exist.

### Funding

| Funder | Grant reference number | Author |
|---|---|---|
| Russian Science Foundation | RSF 20-15-00351 | Yuri B Lebedev |
| Deutsche Forschungsgemeinschaft | Exc2167 | Andre Franke |
| Deutsche Forschungsgemeinschaft | 4096610003 | Andre Franke |
| H2020 European Research Council | COG 724208 | Aleksandra M Walczak |
| Russian Foundation for Basic Research | 19-54-12-011 | Ilgar Z Mamedov |
| Russian Foundation for Basic Research | 18-19-09132 | Ilgar Z Mamedov |
| Ministry of Science and Higher Education | 075-15-2019-1789 | Dmitriy M Chudakov |

The funders had no role in study design, data collection and interpretation, or the decision to submit the work for publication.

### Author contributions

Anastasia A Minervina, Conceptualization, Data curation, Software, Formal analysis, Validation, Investigation, Visualization, Methodology, Writing - original draft, Writing - review and editing; Ekaterina A Komech, Validation, Investigation, Methodology, Writing - review and editing; Aleksei Titov, Validation, Investigation, Visualization, Methodology, Writing - review and editing; Meriem Bensouda Koraichi, Software, Formal analysis, Investigation, Methodology; Elisa Rosati, Resources, Investigation, Methodology, Project administration, Writing - review and editing; Ilgar Z Mamedov, Resources, Supervision, Funding acquisition, Methodology, Project administration, Writing - review and editing; Andre Franke, Resources, Supervision, Funding acquisition, Project administration; Grigory A Efimov, Resources, Supervision, Investigation, Methodology, Writing - review and editing; Dmitriy M Chudakov, Yuri B Lebedev, Conceptualization, Resources, Supervision, Funding acquisition, Methodology, Project administration, Writing - review and editing; Thierry Mora, Conceptualization, Formal analysis, Supervision, Investigation, Methodology, Writing - original draft, Project administration, Writing - review and editing; Aleksandra M Walczak, Conceptualization, Formal analysis, Supervision, Funding acquisition, Investigation, Methodology, Writing - original draft, Project administration, Writing - review and editing; Mikhail V Pogorelyy, Conceptualization, Data curation, Software, Formal analysis, Supervision, Investigation, Methodology, Writing - original draft, Project administration, Writing - review and editing

## Author ORCIDs

Anastasia A Minervina ⬤ https://orcid.org/0000-0001-9884-6351
Elisa Rosati ⬤ http://orcid.org/0000-0002-2635-6422
Dmitriy M Chudakov ⬤ http://orcid.org/0000-0003-0430-790X
Thierry Mora ⬤ http://orcid.org/0000-0002-5456-9361
Aleksandra M Walczak ⬤ http://orcid.org/0000-0002-2686-5702
Yuri B Lebedev ⬤ https://orcid.org/0000-0003-4554-4733
Mikhail V Pogorelyy ⬤ https://orcid.org/0000-0003-0773-1204

## Ethics

Human subjects: All subjects gave written informed consent in accordance with the Declaration of Helsinki. The study protocol was approved by the Pirogov Russian National Research Medical University local ethics committee (#194 granted on March 16, 2020).

## Decision letter and Author response

Decision letter https://doi.org/10.7554/eLife.63502.sa1
Author response https://doi.org/10.7554/eLife.63502.sa2

# Additional files

## Supplementary files

- Supplementary file 1. List of all TCRbeta and TCRalpha libraries produced in this study.
- Supplementary file 2. HLA-typing results for donors M and W.
- Supplementary file 3. List of TCRbeta clonotypes contracting from day 15 to day 85.
- Supplementary file 4. List of TCRalpha clonotypes contracting from day 15 to day 85.
- Supplementary file 5. List of TCRbeta clonotypes expanding from day 15 to day 37.
- Supplementary file 6. List of TCRalpha clonotypes expanding from day 15 to day 37.
- Transparent reporting form

## Data availability

Raw sequencing data are deposited to the Short Read Archive (SRA) accession: PRJNA633317. Resulting repertoires of SARS-CoV-2-reactive clones can be found in SI Tables 3-6 and also accessed from: https://github.com/pogorely/Minervina_COVID (Copy archived at https://archive.softwareheritage.org/swh:1:rev:d5b3953d3739f21a1ccc23114bfffa002e11951e/). Processed TCRalpha and TCRbeta repertoire datasets are available at : https://zenodo.org/record/3835955.

The following datasets were generated:

| Author(s) | Year | Dataset title | Dataset URL | Database and Identifier |
|---|---|---|---|---|
| Minervina AA, Pogorelyy MV, Komech EA, Karnaukhov VK, Bacher P, Rosati E, Franke A, Chudakov DM, Mamedov IZ, Lebedev YB, Mora T, Walczak AM | 2020 | Longitudinal high-throughput TCR repertoire profiling reveals the dynamics of T cell memory formation after mild COVID-19 infection | https://www.ncbi.nlm.nih.gov/bioproject/PRJNA633317/ | NCBI BioProject, PRJNA633317 |
| Minervina AA, Pogorelyy MV, Komech EA, Karnaukhov VK, Bacher P, Rosati E, Franke A, Chudakov DM, Mamedov IZ, | 2020 | Longitudinal high-throughput TCR repertoire profiling reveals the dynamics of T cell memory formation after mild COVID-19 infection | https://zenodo.org/record/3835955 | Zenodo, 10.5281/zenodo.3835955 |

Lebedev YB, Mora T, Walczak AM

The following previously published datasets were used:

| Author(s) | Year | Dataset title | Dataset URL | Database and Identifier |
|---|---|---|---|---|
| Minervina AA, Pogorelyy MV, Komech EA, Karnaukhov VK, Bacher P, Rosati E, Franke A, Chudakov DM, Mamedov IZ, Lebedev YB, Mora T, Walczak AM | 2019 | Comprehensive analysis of antiviral adaptive immunity formation and reactivation down to single cell level | https://www.ncbi.nlm.nih.gov/bioproject/PRJNA577794 | NCBI BioProject, PRJNA577794 |
| Nolan S, Vignali M, Klinger M, Dines J, Kaplan I, Svejnoha E, Craft T, Boland K, Pesesky M, Gittelman RM, Snyder TM, Gooley CJ, Semprini S, Cerchione C, Mazza M, Delmonte OM, Dobbs K, Carreño-Tarragona G, Barrio S, Sambri V, Martinelli G, Goldman J, Heath JR, Notarangelo LD, Carlson JM, Martinez-Lopez J, Robins H | 2020 | A large-scale database of T-cell receptor beta (TCRb) sequences and binding associations from natural and synthetic exposure to SARS-CoV-2 | https://clients.adaptive-biotech.com/pub/covid-2020 | ImmuneAccess, 10.21417/ADPT2020COVID |
| Emerson R, DeWitt W, Vignali M, Gravley J, Hu J, Osborne E, Desmarais C, Klinger M, Carlson C, Hansen J, Rieder M, Robins H | 2017 | Immunosequencing identifies signatures of cytomegalovirus exposure history and HLA-mediated effects on the T-cell repertoire | https://clients.adaptive-biotech.com/pub/emerson-2017-natgen | ImmuneAccess, 10.21417/B7001Z |

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
