## [Decision Letter]

**Acceptance summary:**

This paper uses high throughput longitudinal TCR sequencing to understand the TCR dynamics in two persons infected with SARS-CoV-2. In particular, they find two peaks of T cell clonal expansion at 15 and 37 days post infection. The authors also identify TCR sequence motifs that are likely to be specific to SARS-CoV-2 and thereby show that some T cell clones are present pre-infection suggesting the existence of cross-reactive memory T cells prior to the infection.

**Decision letter after peer review:**

[Editors’ note: the authors submitted for reconsideration following the decision after peer review. What follows is the decision letter after the first round of review.]

Thank you for submitting your work entitled "Longitudinal TCR repertoire profiling reveals the dynamics of T cell memory formation after mild COVID-19" for consideration by *eLife*. Your article has been reviewed by three peer reviewers, and the evaluation has been overseen by a Reviewing Editor and a Senior Editor. The following individual involved in review of your submission has agreed to reveal their identity: Phil Bradley (Reviewer #2).

Our decision has been reached after consultation between the reviewers. Based on these discussions and the individual reviews below, we regret to inform you that your work will not be considered further for publication in *eLife*.

While we found the questions raised in this study interesting, we concluded that the manuscript would not be suitable for *eLife* without substantially more evidence that the clonotypes have TCR specificity for SARS-CoV-2. The individual reviews are appended below.

Reviewer #1:

This work investigates the T cell receptor (TCR) repertoires of 2 individuals diagnosed with mild COVID-19 infection. The authors use high-throughput sequencing of 2 biological replicate samples obtained at each of multiple pre-infection and post-infection timepoints to identify TCRalpha and TCRbeta clonotypes that contract or expand post-infection and to investigate potential reactivation of pre-existing memory cells. This is a potentially interesting work that may provide novel insights into T cell responses to SARS-CoV-2. However, some of the specific details of the various analyses reported are not clear and I have several major concerns about the reported work.

1) The primary concern is the TCR specificity of the clonotypes that were determined to be contracting or expanding post-SARS-CoV-2-infection and therefore identified as responding to or reactive to SARS-CoV-2. There is no verification that these expanding or contracting clonotypes have TCR specificity for SARS-CoV-2. One alternative possibility is that some, maybe even many, of these expanding or contracting clonotypes are bystander-activated T cells with TCRs that are not specific for SARS-CoV-2. Similarly, the clonotypes that were identified as contracting or expanding post-SARS-CoV-2 infection and also detected in the memory pool prior to SARS-CoV-2 infection may not be cross-reactive (i.e. specificity for another infection + SARS-CoV-2), as suggested by the authors, but rather non-SARS-CoV-2-specific bystander-activated memory T cells.

While the dynamics of the T cell populations following SARS-CoV-2 infection may be informative regardless of the mode of activation of the T cells (i.e. TCR-mediated vs. bystander activated), the reported TCR clonotype motifs are only informative if these TCRs have SARS-CoV-2 specificity.

2) Another concern is the substantial variation between the various approaches used to identify the contracting and expanding clonotypes post-infection that are associated with COVID-19 infection. The manuscript text states that the EdgeR and NoiseET approaches for identifying expanding and contracting clonotypes yielded similar results. Figure 1—figure supplement 4A, D suggest that the two approaches yield similar trajectories for the identified expanding and contracting clonotype subsets (i.e. fraction of reactive clonotypes). However, the Venn diagrams in Figure 1—figure supplement 4B, C, E, F show that the two approaches are, in some cases, identifying substantially different subsets of expanding or contracting clonotypes. For example, for Donor M in Figure 1—figure supplement 4F, of the 1044 expanded clonotypes identified by NoiseET, only 478 were also identified by EdgeR.

The text also states that the contracting and expanding clonotypes identified using EdgeR largely overlap/correspond to the clusters 2 and 3 of clonal trajectories yielded using PCA (Figure 1B-E) but no quantitative evidence is provided to support this. Venn diagrams, similar to those in Figure 1—figure supplement 4, could be provided that compare the expanding and contracting clonotypes identified using the three different approaches (i.e. EdgeR, NoiseET, and PCA) as applied to TCRα as well as TCRβ clonotypes.

While these differences between methods may not have significant consequences for some of the reported results (eg. temporal clonal trajectories), these differences raise concerns about the results that depend on specific clonotype sequences (eg. Figure 2D-G, Figure 3—figure supplement 2 and Figure S5D-G that report amino acid motifs for contracting and expanding clonotypes).

Reviewer #2:

This manuscript describes a longitudinal study of TCR repertoires in two individuals with mild COVID-19. TCRα and β repertoires at 4 time points post-infection are used to identify T cell clonotypes likely responding to COVID-19. These responding clones fall into two groups, a set of monotonically contracting clones and a set of clones whose frequencies peak (at day ~37) and then contract. Sequencing of memory populations at two time points and availability of TCR repertoire data from both individuals prior to infection allow the authors to map clonotypes to memory phenotypes and to identify a handful of responding clones that existed in the memory compartment prior to infection. Clusters of sequence-similar clonotypes are identified that suggest focused responses to immunodominant epitopes. This is a succinct and timely study and I have no major concerns, just a few questions/suggestions/typos detailed below.

How unexpected is the TCR clustering evident in Figure 2D-G? For example if the same number of equally high Pgen sequences were selected at random? I wonder whether the authors could run ALICE on just the responding clones (not the full dataset) to assess which neighborhoods are very unlikely to occur by chance.

Could the "computational chain pairing" method of Minervina et al. be applied to this data? If only to try to connect some of the sequence motifs between the α and β chains?

Reviewer #3:

This is a case report analysing TCR repertoire on two individuals with suspected COVID-19 infection. The report shows that a set of TCR sequences expands between days 15 and day 30/37 and another set contract. The amount of expansion/contraction is not clearly shown. Most of these sequences are found in the memory phenotype. A few (especially CD4) are found before immunisation. As the authors point out, the evidence that the TCRs recognise COVID-19 is purely circumstantial. Even if they do, I do not see that this study contributes significantly to understanding either the protective or the pathological immune response to COVID-19.

1) The Abstract is full of unsubstantiated claims. For example "T cell response is a critical part of both individual and herd immunity to SARS-CoV-2 and the eﬃcacy of developed vaccines. " Or "In both donors we identiﬁed SARS-CoV-2-responding CD4^+^ and CD8^+^ T cell clones. We describe characteristic motifs in TCR sequences of COVID- 19-reactive clones, suggesting the existence of immunodominant epitopes." The authors do not identify COVID-19 responding clones; nor do they show any evidence that there are immunodominant epitopes.

2) Figure 1 What does "normalized trajectory of TCR clones in each cluster" mean ? It would be interesting to see the magnitude of the responses. Similarly, I don't really understand the y axis in panels d and e.

3) Figure 3. I don't understand panels a and b. Is this the proportion of contracting TCR sequences which are memory phenotype? If so, what are the rest ? Or are they simply not captured. The figure legend is obscure.

---

## [Author Response]

[Editors’ note: the authors resubmitted a revised version of the paper for consideration. What follows is the authors’ response to the first round of review.]

While we found the questions raised in this study interesting, we concluded that the manuscript would not be suitable for eLife without substantially more evidence that the clonotypes have TCR specificity for SARS-CoV-2. The individual reviews are appended below.

In this resubmission, we now provide such validation using three different approaches:

1) For the HLA-A02 positive donor M, we have acquired TCRbeta and TCRalpha sequences of TCRs binding a HLA-A02 tetramer loaded with a known SARS-CoV-2 peptide. We show that TCRs identified in this assay are independently found in our longitudinal analysis (see Figure 3A in the manuscript).

2) Additionally to our data, a large dataset of SARS-CoV-2 specific TCRs from SARSCoV-2 peptide stimulation experiments of CD8^+^ T cells became available just recently (Snyder et al., 2020, medarxiv). We used this large set of TCR-peptide pairs to further map the specificities of COVID-19-associated TCRs identified by our longitudinal approach. Importantly, the results from this computational mapping agree well with the results we have from analysing HLA-A02 tetramer-specific clones (see Figure 3C in the manuscript). This new analysis also allows us to estimate the fraction of expanding CD8^+^ T cells corresponding to TCRs specific for certain SARS-CoV-2 peptides in vivo.

3) Finally, to validate CD4^+^ clonotypes (points 1 and 2 above only concerned CD8^+^ cells), we used a large dataset of bulk TCRbeta repertoires from 1414 COVID-19 patients reported in the same study (Snyder et al., 2020). We show that the CD4^+^ motifs we report are significantly expanded in COVID-19 patients in comparison to healthy controls (see Figure 4C,D in the manuscript). Moreover, we show that these CD4^+^ motifs are in fact present in the majority of COVID-19 patients with matching HLA-context.

Another point raised by the reviewers was a lack of diagnostic potential of our method. We now show that the motifs identified from the longitudinal analysis are able to identify COVID positive patients.

In this revision we have also added a new timepoint at 3 months for each donor (both bulk and memory repertoires). The analysis of this timepoint provides additional information about the decay of COVID-19 associated T cell clones and their transition to long-lived memory subpopulations on a larger time scale.

Reviewer #1:This work investigates the T cell receptor (TCR) repertoires of 2 individuals diagnosed with mild COVID-19 infection. The authors use high-throughput sequencing of 2 biological replicate samples obtained at each of multiple pre-infection and post-infection timepoints to identify TCRalpha and TCRbeta clonotypes that contract or expand post-infection and to investigate potential reactivation of pre-existing memory cells. This is a potentially interesting work that may provide novel insights into T cell responses to SARS-CoV-2. However, some of the specific details of the various analyses reported are not clear and I have several major concerns about the reported work.1) The primary concern is the TCR specificity of the clonotypes that were determined to be contracting or expanding post-SARS-CoV-2-infection and therefore identified as responding to or reactive to SARS-CoV-2. There is no verification that these expanding or contracting clonotypes have TCR specificity for SARS-CoV-2. One alternative possibility is that some, maybe even many, of these expanding or contracting clonotypes are bystander-activated T cells with TCRs that are not specific for SARS-CoV-2. Similarly, the clonotypes that were identified as contracting or expanding post-SARS-CoV-2 infection and also detected in the memory pool prior to SARS-CoV-2 infection may not be cross-reactive (i.e. specificity for another infection + SARS-CoV-2), as suggested by the authors, but rather non-SARS-CoV-2-specific bystander-activated memory T cells.While the dynamics of the T cell populations following SARS-CoV-2 infection may be informative regardless of the mode of activation of the T cells (i.e. TCR-mediated vs. bystander activated), the reported TCR clonotype motifs are only informative if these TCRs have SARS-CoV-2 specificity.

We thank the reviewer for the valuable comments. We agree that the lack of evidence of SARS-CoV-2 specificity of identified responding clonotypes was the main limitation of the previous version of the manuscript. In the revised version we provide both experimental and computational proof of the SARS-CoV-2 specificity of reported clonotypes. Donor M from the current study also participated in the study by Shomuradova et al. 2020 (as donor p1434), where T cells were sorted using a HLA-A02 tetramer loaded with a SARS-CoV-2 epitope from the spike protein. Sequences of TCRs specific for this SARS-CoV-2 epitope (YLQPRTFLL) were deposited to VDJdb after we shared our original preprint. We now use these sequences to prove that our approach is capable of identifying clonotypes specific to a certain SARS-CoV-2 epitope by clonal dynamics only, see Figure 3A. We show that a) tetramer-specific TCRs show temporal dynamics corresponding to the contracting clones’ group (monotonically decreasing in frequency after day 15 post-infection). b) Our approach identifies almost all tetramerspecific TCRs which are abundant on day 15. c) some groups of highly similar TCR sequences of contracting clones in TCRbeta and TCRalpha repertoire correspond to tetramer-specific clones.

We next compared our CD8^+^ TCRbeta sequences with a recently published large set of TCRbetas from T cells stimulated with SARS-CoV-2 peptide pools (MIRA assay from Snyder et al., 2020, medarxiv). This allowed us to map specificities of our TCRbeta sequences to certain peptides/or pools of overlapping peptides from the MIRA assay. Importantly, this procedure correctly mapped the specificity of TCRbetas from tetramerspecific clones described above. TCR specific for one of MIRA epitopes (NQKLIANQF) were found in the pre-existing memory of donor M. Interestingly this epitope is just one mismatch away from the homologous peptide (NQKLIAN**A**F) of common cold betacoronaviruses, suggesting potential cross-reactivity between these two epitopes.

Finally, we prove SARS-CoV-2 association of discovered CD4^+^ TCRbeta contracting motifs in both donors using a large cohort of COVID-19 patients from ref. Snyder et al., 2020. We show that occurrence of TCRbetas from the largest sequence similarity groups of both donors is predictive of COVID-19 positivity in donors with matching HLA-context.

1) New Figure 3A shows longitudinal tracking of A*02:01-YLQPRTFLL tetramer clones (Figure 3—figure supplement 1 for TCRalpha).

2) Occurence of tetramer clones in clusters of similar contracted TCRbetas and TCRalpha is shown on Figure 3B and c respectively.

3) Mapping of clusters of the similar CD8 TCRbetas to SARS-CoV-2 specific peptides from MIRA experiment is shown on Figure 3C as shaded circles.

4) Occurrence of main CD4^+^ motifs from contracting clones of both donors in large cohorts of COVID-19 patients and pre-pandemic controls is shown on Figure 4C,D.

5) Separate columns in Supplementary file 3 shows if clones were detected in the tetramer enrichment experiment from Shomuradova et al., as well as specificity for peptide/peptide pool obtained by comparison with MIRA database.

6) Both Results and Materials and methods sections are updated to describe new results.

7) We note that none of these methods proved SARS-CoV-2 specificity of clones with a peak on day 37 (either because they are not specific for SARS-CoV-2, or because our validation methods are more suitable for CD8 clones). We discuss this issue in the Discussion.

2) Another concern is the substantial variation between the various approaches used to identify the contracting and expanding clonotypes post-infection that are associated with COVID-19 infection. The manuscript text states that the EdgeR and NoiseET approaches for identifying expanding and contracting clonotypes yielded similar results. Figure 1—figure supplement 4A, D suggest that the two approaches yield similar trajectories for the identified expanding and contracting clonotype subsets (i.e. fraction of reactive clonotypes). However, the Venn diagrams in Figure 1—figure supplement 4B, C, E, F show that the two approaches are, in some cases, identifying substantially different subsets of expanding or contracting clonotypes. For example, for Donor M in Figure 1—figure supplement 4F, of the 1044 expanded clonotypes identified by NoiseET, only 478 were also identified by EdgeR.The text also states that the contracting and expanding clonotypes identified using EdgeR largely overlap/correspond to the clusters 2 and 3 of clonal trajectories yielded using PCA (Figure 1B-E) but no quantitative evidence is provided to support this. Venn diagrams, similar to those in Figure 1—figure supplement 4, could be provided that compare the expanding and contracting clonotypes identified using the three different approaches (i.e. EdgeR, NoiseET, and PCA) as applied to TCRα as well as TCRβ clonotypes.While these differences between methods may not have significant consequences for some of the reported results (eg. temporal clonal trajectories), these differences raise concerns about the results that depend on specific clonotype sequences (eg. Figure 2D-G, Figure 3—figure supplement 2 and Figure S5 D-G that report amino acid motifs for contracting and expanding clonotypes).

To ensure that all of our results are reproducible with both the NOISET and edgeR approaches, we redid all the analysis and now call a clonotype “contracting” or “expanded” if it was called significant by both approaches simultaneously. Hence we show that active clonal dynamics, participation of the pre-existing memory in the response and SARSCoV-2 TCR sequence motifs we report in our initial preprint are successfully captured by both edgeR and NoisET noise models.

All figures and values in the manuscript are updated with significantly contracting and expanding clonotype groups identified with both models.

We used PCA as an exploratory data analysis technique to identify general patterns in clonotype dynamics and inform the selection of timepoints for more precise analysis with edgeR/NoisET. We do not use PCA to identify significantly contracted/expanded clones, as PCA does not take noise into account.

The overlap between clusters identified by clustering of the principal components and significant clonotypes called by both edgeR and NoisET simultaneously is now shown on Figure 1—figure supplement 5.

We also added different shapes in Figure 1BC and Figure 1—figure supplement 3A-B to indicate clonotypes called as expanding/contracting by both edgeR and NoisET.

Reviewer #2:This manuscript describes a longitudinal study of TCR repertoires in two individuals with mild COVID-19. TCRα and β repertoires at 4 time points post-infection are used to identify T cell clonotypes likely responding to COVID-19. These responding clones fall into two groups, a set of monotonically contracting clones and a set of clones whose frequencies peak (at day ~37) and then contract. Sequencing of memory populations at two time points and availability of TCR repertoire data from both individuals prior to infection allow the authors to map clonotypes to memory phenotypes and to identify a handful of responding clones that existed in the memory compartment prior to infection. Clusters of sequence-similar clonotypes are identified that suggest focused responses to immunodominant epitopes. This is a succinct and timely study and I have no major concerns, just a few questions/suggestions/typos detailed below.How unexpected is the TCR clustering evident in Figure 2D-G? For example if the same number of equally high Pgen sequences were selected at random? I wonder whether the authors could run ALICE on just the responding clones (not the full dataset) to assess which neighborhoods are very unlikely to occur by chance.

We show that clustering events are indeed unlikely to occur by chance. This can be assessed by creating a random sample of sequences from the repertoire of the same size as the responding subset, and by comparing the average number of edges in their similarity networks. That number is significantly lower in the random subset than in the responding subset (as expected, with exception of expanding CD8^+^ clones group, there is no strong motif there).

We added new Figure 3D and Figure 4—figure supplement 1A showing the distribution of the number of edges expected under a random sampling assumption.

Could the "computational chain pairing" method of Minervina et al. be applied to this data? If only to try to connect some of the sequence motifs between the α and β chains?

We thank the reviewer for this suggestion. We applied a pairing method from Minervina et al., and found that most clusters do not show consistent pairing with each other. However, we identified two cases in CD8^+^ contracting clones with repeated cluster-to cluster pairings.

We have added this information in the Materials and methods. The sequence of the predicted α chain for each TCRbeta clone is given in Supplementary file 3 and Supplementary file 5.

Reviewer #3:This is a case report analysing TCR repertoire on two individuals with suspected COVID-19 infection. The report shows that a set of TCR sequences expands between days 15 and day 30/37 and another set contract. The amount of expansion/contraction is not clearly shown. Most of these sequences are found in the memory phenotype. A few (especially CD4) are found before immunisation. As the authors point out, the evidence that the TCRs recognise COVID-19 is purely circumstantial. Even if they do, I do not see that this study contributes significantly to understanding either the protective or the pathological immune response to COVID-19.1) The Abstract is full of unsubstantiated claims. For example "T cell response is a critical part of both individual and herd immunity to SARS-CoV-2 and the eﬃcacy of developed vaccines. " Or "In both donors we identiﬁed SARS-CoV-2-responding CD4^+^ and CD8^+^ T cell clones. We describe characteristic motifs in TCR sequences of COVID- 19-reactive clones, suggesting the existence of immunodominant epitopes." The authors do not identify COVID-19 responding clones; nor do they show any evidence that there are immunodominant epitopes.

We agree that evidence for these claims was suggestive in the initial preprint. However we believe we now provide strong evidence for these points in the revised version:

1) We provide evidence that clones identified by our longitudinal analysis are indeed COVID-19 specific with 3 different methods: by the sequencing of TCRbeta and TCRalpha of T cell clones stained with HLA*A02-tetramers loaded with YLQPRTFLL (peptide from SARS-CoV-2 spike, see Figure 3A); by matching of TCRbeta sequences to a large recently published dataset of SARS-CoV-2 reactive CD8^+^ T cells (MIRA peptide assay, Snyder et al., 2020, see Figure 3C); and by showing preferential occurrence of both CD8 and CD4 T cell motifs in a large TCRbeta dataset of 1414 COVID-19 patients (Snyder et al., 2020), Figures 4CD.

2) In the original preprint we showed that T cell clones with active dynamics after COVID-19 frequently feature highly similar TCRbeta/TCRalpha chains. It was previously shown by multiple groups that similar TCR sequences frequently recognise the same epitope. We hypothesized that clusters of highly similar TCRs, corresponding to the large fraction of expanding cells, recognise certain immunodominant epitopes.

By sequencing TCRs from tetramer-binding sorted cells, and by searching for TCRbeta clones in databases of epitope-specific TCRs, we were able to confirm this suggestion. The new Figure 3E shows the fraction of the CD8^+^ response mapped to certain peptides, showing that the CD8^+^ T cells responding to the spike-derived HLA-B*15:01 restricted NQKLIANQF epitope corresponds to 21% of CD8^+^ response at its peak. Another prominent epitope is HLA-A*02:01-YLQPRTFLL, which corresponded to 4% of CD8^+^ response at the peak. The occurrence of the described CD4 motifs in a substantial part of HLA-matched COVID-19 patients from Snyder et al., 2020 also points towards the existence of immunodominant CD4 epitopes in the SARS-CoV-2 genome, which elicit a response in the majority of COVID-19 patients with matching HLA-context.

1) We modified the Abstract and reformulated the claims pointed out by the reviewer to be more specific.

2) We added additional analyses proving the COVID-19 specificity of the TCRs we identified and mapping them to certain epitopes:

a) New Figure 3A shows longitudinal tracking of A*02:01-YLQPRTFLL tetramer specific clones (Figure 3—figure supplement 1 for TCRalpha).

b) The occurrence of tetramer-positive clones in clusters of similar contracted TCRalpha and TCRbeta clonotypes is shown in Figure 3B and C.

c) The mapping of TCRbeta clusters to SARS-CoV-2 specific peptides from the MIRA experiment is shown in Figure 3C (shaded circles).

d) The occurrence of the main CD4^+^ motifs of contracting clones from both donors in large cohorts of COVID-19 patients and the pre-pandemic control cohort is shown in Figure 4C, D.

e) Separate columns in Supplementary file 3 report if clones were detected in the tetramer enrichment experiment from Shomuradova et al., as well as the specificity for peptides from the MIRA database.

f) Both the Results and Materials and methods sections are updated to describe the new analyses.

2) Figure 1 What does "normalized trajectory of TCR clones in each cluster" mean ? It would be interesting to see the magnitude of the responses. Similarly, I don't really understand the y axis in panels d and e.

1) We reformulated and expanded Figure 1’s caption to make it clear that clonal frequencies are normalized by the maximum value for each clone in Figure 1B-C, and that Figure 1B-C’s left panels show the average of such normalized frequencies within each cluster (and thus show the dynamical pattern of an average clone in that cluster).

2) We now discuss the magnitude of the responses in the text.

3) We reformulated the Figure 1D-E caption to explain the y-axis more clearly: it is the sum of the frequencies of all clonotypes detected as “contracting” (D) and “expanding” (E).

3) Figure 3. I don't understand panels a and b. Is this the proportion of contracting TCR sequences which are memory phenotype? If so, what are the rest ? Or are they simply not captured. The figure legend is obscure.

Yes, this is correct. These data are now shown in Figure 2, and we have expanded its caption to explain it more clearly. We profile PBMC repertoires more deeply than repertoires of FACS sorted memory cells, and thus indeed low-abundant responding clones are captured in PBMC, but not in memory subpopulations. We have added a discussion of this sampling issue in the text.